# Functional Living Skills in Patients with Major Neurocognitive Disorder Due to Degenerative or Non-Degenerative Conditions: Effectiveness of a Non-Immersive Virtual Reality Training

**DOI:** 10.3390/s23041896

**Published:** 2023-02-08

**Authors:** Simonetta Panerai, Valentina Catania, Francesco Rundo, Domenica Tasca, Sabrina Musso, Claudio Babiloni, Giuseppina Prestianni, Stefano Muratore, Raffaele Ferri

**Affiliations:** 1Unit of Psychology I.C., Oasi Research Institute-IRCCS, 94018 Troina, Italy; 2Unit of Bioinformatics and Statistics, Oasi Research Institute-IRCCS, 94018 Troina, Italy; 3Department of Physiology and Pharmacology “V. Erspamer”, Sapienza University of Rome, 00185 Rome, Italy; 4Unit of Neurology I.C., Oasi Research Institute-IRCCS, 94018 Troina, Italy

**Keywords:** virtual reality, neurocognitive disorders, degenerative dementia, non-degenerative dementia, functional living skills

## Abstract

Virtual reality has gained attention as an effective tool for cognitive, motor, and daily activity rehabilitation in patients with major neurocognitive disorder (M-NCD). The first objective of this study was to check for differences between M-NCD caused by degenerative and non-degenerative conditions (DC and NDC, respectively) in terms of relearning four functional living skills (FLSs), by means of a non-immersive virtual reality training (VRT). The second purpose was to verify whether spontaneous transfer from the virtual environment to the real environment occurred. Four FLS apps were developed in our institute (Information, Suitcase, Medicine, and Supermarket). A nonrandomized interventional study was carried out, comparing experimental and control groups (EG and CG, respectively). The study included three phases: in vivo test at T1; VRT at T2 only for EG; in vivo test at T3. During the in vivo test, the four FLSs were assessed in their natural environments. Both EG-DC and EG-NDC significantly improved in all of the VRT variable scores (the EG-NDC group seemed to show better outcomes than the EG-DC group). Moderate-to-high satisfaction with the VRT was reported. EG-DC and EG-NDC also enhanced their performances in the in vivo test. No statistically significant differences between them were found. CG-DC and CG-NDC improved only in the execution time of Information in the in vivo test. These findings confirm the ecological validity of VRT for FLSs.

## 1. Introduction

Major neurocognitive disorder (M-NCD) [1], or dementia, is an umbrella term including a wide range of chronic conditions, characterized by a significant cognitive decline in one or more cognitive domains, which impacts the functional living skills (FLSs) that are essential for independent living. The loss of independence in M-NCDs is associated with a worse quality of life, decreased self-esteem, increased risk of caregiver distress, and institutionalization [2,3]. M-NCDs include some etiological subtypes, such as neurodegenerative (e.g., Alzheimer’s disease, frontotemporal lobar degeneration, Parkinson’s disease) and non-neurodegenerative conditions (e.g., vascular disease, traumatic brain injury, brain tumors). Neurodegenerative dementia seems to be more frequent in the elderly, whereas non-degenerative dementia tends to occur at a younger age, in adults or younger adults [4]. M-NCDs require both pharmacological and non-pharmacological longitudinal care. Since pharmacological treatments to date have not proven to be decisive, interest has focused on non-pharmacological (NP) approaches [5] in order to modify the disease course. NP treatments are preferred in order to deal with the behavioral and emotional symptoms of dementia (e.g., music, validation therapy, and sensory stimulation), as well as to maintain FLSs and cognitive skills (e.g., exercises, cognitive stimulation, and reminiscence) [6,7]. Among the clinical tools for assessment and rehabilitation, virtual reality (VR) is increasingly affirming its effectiveness. It has a long history, beginning in the 1960s in the graphic field and then extending to many domains, including physical health, mental disorders, neuroscience, and neuropsychology [8,9,10]. In the neuropsychological field, VR has been spotlighted as an innovative tool; as an alternative or complement to the traditional approaches, it can measure executive functions and other cognitive abilities not only through virtual versions of paper-and-pencil tests, but also—and especially—in everyday life simulations [10]. Indeed, a specific relationship between executive functions and activities of daily living has been found [11]. So far, most of the studies using VR have concerned assessment and validation [10]; however, especially in the last decade, VR has been increasingly implicated in the rehabilitation field [8] and has proven to be a suitable tool to treat executive and other cognitive functions simulating daily life contexts. Previous studies have mainly concerned inhibition, working memory, shifting, and planning, as well as attention, and visuospatial and motor functioning, in virtual environments such as classrooms, markets, and apartments [10].

VR has gained attention as an effective tool for cognitive, motor, and daily activity rehabilitation in patients with M-NCDs or in the prodromal conditions of dementia. It seems to produce positive effects in a short time, as well as to enhance brain neuroplasticity [12]. Furthermore, it seems to be well accepted by people with Alzheimer’s disease, frontotemporal dementia, and mild cognitive impairment, and to be generally preferred to traditional approaches [5], emerging as a valid tool for improving patients’ involvement and adherence to motor and cognitive rehabilitation.

VR consists of a three-dimensional (3D) computer simulation of a real environment. It allows users to feel “to be there” and to interact with and within the specific simulated place. Three types of immersive systems have been described: low-immersion, in which the patient interacts with the simulated environment through a PC monitor (touchscreen or keyboard and mouse), with or without additional sounds; semi-immersive, using more complex interactive devices (such as haptic gloves and balance platforms); and full-immersion, generally using head-mounted displays connected with any computer or video source, or the cave automatic virtual environment, which projects images on the wall, floor, or ceiling. The full-immersion systems also integrate sensory stimuli that make the simulated environment even more realistic [13]. A comparison between the experiences of young adult and senior individuals in using two versions of the virtual supermarket shopping task (a full-immersion version with a head-mounted display and a low-immersion version with a desktop) showed no differences in the preference of one platform over the other, although immersive VR resulted in more fatigue in both groups; however, seniors performed better with the desktop system [14].

VR applications need to meet specific needs, especially those for motor and cognitive rehabilitation. They can be personalized according to the strengths and weaknesses of the patients. A recent review showed positive effects on memory, global cognition, and balance in people with mild cognitive impairment or dementia [13]. Recent studies comparing immersive VR and conventional rehabilitation in Parkinson’s disease found greater improvements in dynamic balance and gait with VR than with conventional rehabilitation [15]. Low-immersion VR seems to be highly accepted by older people, due to minimal sickness symptoms; it also shows positive effects in the rehabilitation of the main geriatric disorders, especially in improving visuospatial abilities, executive functioning, and gait, as well as decreasing fall rates in Parkinson’s disease [16,17] and increasing balance in stroke patients [16]. A recent interesting study [18] showed more improvement in balance, reduced risk of falls, and enhanced emotional and mental health status in patients with Parkinson’s disease, after non-immersive VR exergames, than in controls, who received traditional rehabilitation. Therefore, the non-immersive VR appeared to be useful for training different domains simultaneously, such as motor, cognitive, and emotional domains.

Some authors have emphasized that the interventions for people with dementia should be guided by the effort to maintain FLSs for as long as possible [4], in order to support independence and integration in the community. These essential skills include complex activities of daily life, such as financial management, shopping, driving, home management, and medication management. The link between neuropsychological functions and FLSs has been investigated, and memory, executive functions, and language seem to be involved [19]. FLSs progressively deteriorate as a result of the cognitive decline in patients with M-NCDs, representing a great burden for caregivers and the family economy [19]. VR is considered to be a promising tool not only for improving motor and cognitive abilities, but also for the training of FLSs [16,20,21]; indeed, several studies have found a significant relationship between virtual and real tasks [21], improvements in functional living [22,23,24,25,26,27,28], and cognitive skills [27], as well as in cases of apathy [26], maintenance of FLSs over time [22], and spontaneous transfer in the natural environments of FLSs relearned in VR environments [29].

The objective of the present study was to compare the effectiveness of low-immersion VR training, focused on four FLSs (namely, to provide information, to take medicines, to prepare a suitcase, and to shop at the supermarket), in patients with M-NCDs with different etiologies, i.e., degenerative or non-degenerative conditions (DC and NDC, respectively). In a previous study [29], we found that VR training (VRT) produced improvements in M-NCD patients (i.e., increased number of successes and shorter execution time), as well as a statistically significant difference between patients and controls in performing FLSs in the natural setting. The present study specifically aimed to verify—by means of comparisons within groups, as well as between trained groups and non-trained controls—whether there were differences between M-NCD due to DC or NDC in terms of (a) relearning the four FLSs by means of the non-immersive VRT, and (b) the spontaneous skill transfer from the virtual environment to the real environment. The satisfaction level with the technological system was also investigated in both DC and NDC conditions.

## 2. Materials and Methods

### 2.1. Participants

Participants were recruited by the Unit of Neurorehabilitation of the Brain Aging Department, Oasi Research Institute-IRCCS, Troina (Italy), from November 2018 to November 2022. Inclusion and exclusion criteria were applied. Inclusion criteria were as follows: (a) diagnosis of M-NCD, according to the DSM-5 criteria [1]; (b) Mini-Mental State Examination (MMSE) [30] 10 ≥ scores ≤ 24; (c) loss of one or more instrumental activities of daily living (IADLs) [31]; (d) no serious impairments in sight, hearing, or using the dominant upper limb; (e) maintenance of verbal communication skills; (f) sufficient reading skills. Exclusion criteria were as follows: (a) prodromal phase of dementia; (b) severe degree of dementia; (c) aphasia; (d) severe sight or hearing impairments, or impossibility of using the dominant upper limb; (e) no reading skills. Patients who gave their informed consent were assigned to the VRT experimental group (EG; N = 40, of whom N = 19 with M-NCD due to DC, and N = 21 with M-NCD due to NDC); patients who did not accept VRT were assigned to the control group (CG; N = 27). Patients with M-NCD due to DC in the EG included the following etiological subtypes [1]: N = 12 Alzheimer’s disease, N = 4 frontotemporal lobar degeneration, N = 2 cortical basal degeneration, N = 1 Parkinson’s disease. Patients with M-NCD due to NDC in the EG included the following subtypes [1]: N = 17 vascular disease, N = 1 traumatic brain injury, N = 3 another medical condition. The CG included the following etiological subtypes [1]: N = 10 Alzheimer’s disease, N = 5 frontotemporal lobar degeneration, N = 1 Parkinson’s disease among DC; N = 5 vascular disease, N = 2 traumatic brain injury, N = 4 another medical condition among NDC. The characteristics of the sample are reported in Table 1.

### 2.2. Description of the System and Apps

From a technical point of view, the system is based on a classic client–server architecture. Through a DB and REST APIs, the server allows the management of user and patient profiling, rehabilitation protocols, and data exchange with apps. A web service was been created in NodeJs that exchanges data in JSON format and stores them in the DB, created in PostgreSQL. Furthermore, to guarantee data security, the web service APIs are accessible only and exclusively through an authentication and authorization system.

Web applications (by React) and desktop applications (by Visual Studio) were also developed, through which the rehabilitation protocols can be created and configured, making it possible to define the number of sessions, how many and which tasks to carry out, and on which device. Finally, using the Unity3D graphics engine (C# language), several apps were developed that could be used on different devices and allow the execution and saving of the rehabilitation activity results. 

In this study, four apps were used, each referring to a specific area of the activities of daily living identified by the American Association of Occupational Therapy [32]: to provide information (or Information) refers to the communication area; to take medicines (or Medicines) refers to health management; to prepare a suitcase (or Suitcase) refers to home management; and to shop at the supermarket (or Supermarket) refers to shopping abilities. The design of all of these apps is based on the applied principles of behavioral psychology. Applied behavior analysis (ABA) procedures have shown clear evidence of effectiveness in the rehabilitation of children and adults [33,34], and the joint use of ABA and VR has shown promising results [35,36]. 

All apps developed in our institute include a combination of behavior analytic components, such as (a) antecedent interventions—pre-training is provided on the use of the VR system, and each app starts with written and verbal instructions, followed by the possibility of watching a demonstration video (for Suitcase and Supermarket); (b) verbal reinforcement (e.g., “well done”, or “congratulations”) after correct responses; (c) corrective feedback after incorrect responses, also providing the least-to-most prompting clue (with a maximum of three clues); and (d) task analysis and total task chaining (for Suitcase and Supermarket). 

Three of the apps include a single 3D scene; only Supermarket includes three different 3D scenes. Information and Medicines require the patients to respond by tapping an item in the scene; Suitcase and Supermarket require the patients to drag items from one point to another in the scene. 

Some other details of the apps are described below:Information: 30 questions about personal, general, family, and spatiotemporal information appear on the screen in verbal and written form simultaneously; the patient is asked to choose the answer from a multiple-choice menu by touching one of the options on the screen.Medicines: The task requires the patient to indicate the time of the day at which to take five different types of drugs. In each session, 10 verbal requests are randomly made, and the patient has to touch one of the five medicine boxes on the screen. A visual reminder is always available.Suitcase: 10 items (clothing and accessories) must be packed in a suitcase for a weekend out. The task ends with the closing of the suitcase. There are two versions of the app—one for women and one for men.Supermarket: The activity requires the purchase of 5 items, presented in a shopping list that remains available upon request. The shopping list changes randomly in different sessions. In the first scene, representing a kitchen, the patient takes the shopping list and the wallet with money; in the second scene, they take the products from the supermarket shelf and put them in the cart; in the third scene, at the checkout, they put the products on the conveyor belt, pay, and exit the shop.

### 2.3. Procedures

A nonrandomized interventional comparison study was carried out, including pre-treatment assessment (T1), treatment (T2), and post-treatment assessment (T3).

Pre-treatment assessment (T1): In the pre-treatment phase, all of the patients (M-NCD due to DC or NDC in both EG and CG) were assessed by means of a comprehensive neuropsychological battery, administered by a clinical psychologist who was blinded to the aims of the study, in order to evaluate global mental functioning, executive functioning, memory, and daily living activities. The battery included the MMSE [30], Raven’s Colored Progressive Matrices [37], digit and visuospatial span [38], Rey’s 15 words—immediate and delayed recall— [39], Frontal Assessment Battery [40], activities of daily living [41], and IADL [31]. Table 1 shows the median scores obtained in the neuropsychological battery by the groups. Then, the first in vivo tests were administered in real environments arranged in our institute, in order to evaluate how patients performed the four FLSs (to give information, to take medicines, to prepare a suitcase, and to shop at a supermarket) in a natural setting. The in vivo tests were administered by a behavioral psychologist, using a task analysis form for each activity. The behavioral psychologist did not know the groups to which the different participants had been assigned. No clues or reinforcement were provided, and the number of correct responses and total execution times were recorded. Finally, a digital literacy questionnaire was administered to the EG participants. Table 1 shows the median scores obtained on the in vivo tests by both the EG and CG, along with the median number of technological devices used by the EG participants.Treatment (T2): At T2, both EG-DC and EG-NDC were administered the VRT, with five sessions per week over a period of two weeks. We chose to provide only 10 sessions, based on the results of our previous study [29] in which the most significant results were obtained between the 7th and 10th sessions. VRT was added to the usual cognitive stimulation carried out in the neurorehabilitation department of our institute [2]. As stated above, before VRT, two or three test sessions were conducted for familiarization with the touchscreen and the dragging movement. The test app only involved moving a ball from one point on the screen to another. During the VRT, the patients were required to independently manage their device and tasks; nevertheless, the psychologist was always present in the room for general monitoring of the session’s progress and could intervene in case of system malfunction. The variable scores were automatically saved on the server, namely, total execution time, number of correct responses, number of errors, number of missing responses (i.e., participants not answering within 10 s), and number of clues provided to obtain the response. A video demo is available in the Appendix A.Post-treatment assessment (T3): At T3, a second administration of the in vivo test was repeated for the EG (both DC and NDC) and the CG (both DC and NDC). A satisfaction questionnaire arranged in our institute was administered only to the participants who performed the VRT. It included 14 questions: part I (8 questions) focused on the system usability and satisfaction, while part II (6 questions) focused on the technological problems and the negative feelings and symptoms. Single scores ranged from 0 = low to 2 = high. Part I total scores ranging from 0 to 4 were considered to indicate low satisfaction; scores from 5 to 8 indicated low-to-moderate satisfaction; scores from 9 to 12 indicated moderate-to-high satisfaction; and scores from 13 to 16 indicated high satisfaction. Part II total scores ranging from 0 to 3 indicated a low level of problems; scores from 4 to 6 indicated low-to-moderate problems; scores from 7 to 9 indicated moderate-to-high problems; and scores from 10 to 12 indicated a high level of problems.

### 2.4. Statistical Analysis

The recorded data did not show a normal distribution shape (asymmetry and kurtosis calculations). For this reason, and since the groups were small, nonparametric statistics were used. The comparisons between the EG (DC and NDC) and CG (DC and NDC) were carried out by means of Fisher’s test for dementia etiology and sex (2 × 2 contingency table), and with the chi-squared test for the dementia severity level (4 × 3 contingency table). All of the other comparisons between the four groups were performed using the Kruskal–Wallis test. When a statistically significant difference was found, a post hoc test was conducted with the Mann–Whitney U test and Bonferroni correction (*p*-value was set to 0.012). For the in vivo tests, the intragroup comparisons were performed by using the Wilcoxon matched-pairs test followed by the Bonferroni correction (setting the *p*-value to 0.006). The Friedman test for repeated measures was used in order to compare the 10 VRT sessions carried out with the EG (both DC and NDC), together with the Kendall’s W coefficient of concordance. In order to assess the differences in the session progression between EG-DC and EG-NDC, the 1st, 5th, and 10th sessions were compared by means of the Mann–Whitney U test. The r effect sizes were also calculated (r = z/$\sqrt{N}$; 0.1= small effect size; 0.3 = medium effect size; 0.5 = large effect size). 

## 3. Results

Table 1 shows the results obtained from the baseline comparisons between the EG (DC and NDC) and the CG (DC and NDC), scores from the comprehensive neuropsychological battery, and scores obtained in the first in vivo test (number of correct responses and total execution times). Statistically significant differences between the four groups were found in chronological age and in the number of correct responses in the Suitcase task. The subsequent post hoc test confirmed a significant difference in chronological age only in the CG (i.e., older age in the DC than in the NDC group), which might be expected since neurodegenerative dementias are more frequent in the elderly [4]. Concerning the correct responses in the Suitcase task, a significant difference was confirmed at the post hoc test only between EG-DC and CG-DC. No significant differences emerged from any of the other comparisons; therefore, we considered the four groups to be comparable. As for the use of technological devices, the median scores indicated that the EG participants were familiar with one device (generally a cell phone) in a high percentage of both DC and NDC (about 53% and 48%, respectively); two devices were used by about 16% of DC and 33% of NDC individuals; approximately 20% in both groups used more than two devices; only 10% of DC individuals did not use any devices.

Table 2 shows the results obtained when comparing the differences in scores between the second (T3) and first (T1) in vivo tests. From the Kruskal–Wallis test, no statistically significant differences between the four groups were found in terms of total execution times. Conversely, statistically significant differences were found in the number of correct responses. The post hoc test showed no differences between DC and NDC, nor between the EG and CG. Statistically significant differences were found in the Information, Suitcase, and Supermarket tasks between the degenerative conditions (EG vs. CG), and in the Information and Suitcase tasks between the non-degenerative conditions (EG vs. CG). In the Medicine task, statistically significant differences were found when comparing both DC and NDC to EG and CG, but they did not reach the *p* threshold set after the Bonferroni correction. The intragroup comparisons (Wilcoxon matched-pairs test) showed significant improvements for EG-DC in the number of correct responses in all of the FLSs, as well as in the total execution time for Information. The EG-NDC showed improvements in correct responses in three of the four FLSs tested in the natural environment (i.e., Information, Suitcase, and Supermarket), as well as in the total execution time for Information and Medicines. Only one difference was found in the CG-DC, namely, in the total time for Information. No statistically significant differences were found in the CG-NDC.

The differences across the 10 VRT sessions are reported in Table 3. Both EG-DC and EG-NDC showed statistically significant improvements in almost all of the parameters taken into consideration (Friedman test for repeated measures). Only the number of errors did not show statistically significant differences, except for the Supermarket app in the EG-NDC. 

The Kendall’s W coefficient of concordance was small-to-moderate in correct responses, missing responses, and clues in both groups, while it was moderate-to-high in total execution times in both groups. In general, the Kendall’s W coefficients of concordance values were greater and the *p*-values were smaller in the EG-NDC than in the EG-DC.

Figure 1 shows the progression of the median values of correct responses, missing responses, clues, and total execution time, across the 10 VRT sessions. The number of errors was not considered, as it was almost always non-significant according to the Friedman test for repeated measures. As can be seen from the graphs, the progression across the sessions was similar in the two groups, but the EG-NDC showed better performances than the EG-DC. 

The comparisons between EG-DC and EG-NDC in the 1st, 5th, and 10th sessions of the VRT, carried out by means of the Mann–Whitney U test (Table 4), showed statistically significant differences in the correct responses for Suitcase in the 10th session, Medicines in the 5th session, and Supermarket in both the 5th and 10th sessions; in missing responses for Supermarket in the 5th session, Information in the 10th session, and in Medicine in the 1st session; and in numbers of clues for Medicine in the 5th session.

As for the satisfaction questionnaire, the median score obtained in Part I by the EG-DC was 10 (interquartile range 7.25–12.75), while for the EG-NDC it was 11 (interquartile range 8–13.25), indicating a moderate-to-high satisfaction. The score obtained in Part II by the EG-DC was 0 (interquartile range 0–0.75), and for the EG-NDC it was 0 (interquartile range 0–0.25), indicating a low level of problems with the technological system.

## 4. Discussion

In our previous study [29], the effects of a non-immersive VRT on FLSs in people with dementia were investigated by comparing trained and non-trained individuals with M-NCDs. The results were encouraging, as the trained group showed statistically significant improvements both in performing VR tasks and in performing the corresponding daily living activities in the natural environment (differences between the second and first in vivo tests); in contrast, the non-trained group did not show any improvement in the in vivo test. Our work has continued by enrolling new patients and developing two new FLS apps, which are yet to be tested and are not reported here. Our previous study involved patients with M-NCDs with different etiologies; this did not allow us to generalize the results to different types of dementia. Consequently, in the present study, two clinical group types were included (namely, degenerative and non-degenerative dementias) to check for any differences in outcomes—in particular, (a) whether there were differences in performing VRT sessions between the two treated groups (EG-DC and EG-NDC), and (b) whether after VRT the treated groups showed differences in comparison to controls, when performing the FLS tasks in the natural environment. 

To the best of our knowledge, no previous study has compared the outcomes of VR-based FLS training in patients with M-NCD due to DC or NDC. Recent studies and reviews on VR-based rehabilitation have shown promising results for some cognitive functions in neurodegenerative conditions (especially executive and visuospatial abilities), as well as in post-stroke cognitive impairments [42,43] for verbal memory. It also appears that cognitive improvements after VRT have a positive impact on daily living skills [43], just as VRT for daily living skills has a positive impact on global cognition [44].

With reference to the first objective of the present study, both EG-DC and EG-NDC improved in all of the variable VRT scores—correct responses, missing responses, number of clues, and total execution time—except for the number of errors (Table 3). Therefore, in both groups, the increase in correct responses could probably be related to the decrease in the number of missing responses (and not just to the decrease in errors). Our sample, especially at the beginning of VRT (and especially in the EG-DC), showed more missing responses than errors. These data might be related both to the slower elaboration of the responses (which exceeded the limit of 10 s) and to the uncertainty in responding—typical characteristics of the dementia conditions. During the training, the patients gained greater speed and confidence with the system, so that the number of missing responses decreased while the proportion of correct responses increased. On the other hand, the number of errors always seemed to fluctuate within the same range for each app; in very few cases, 0 errors were made in the last two or three sessions—especially in the EG-DC. The EG-NDC seemed to show better outcomes than the EG-DC. This was confirmed by the comparisons between the EG-DC and EG-NDC in the 1st, 5th, and 10th sessions, which showed some statistically significant differences—primarily in correct responses for Supermarket, Suitcase, and Medicines, followed missing responses for Information, Medicine, and Supermarket, and finally by the number of clues in Medicines, with better performance in EG-NDC than in EG-DC. It seems, therefore, that the EG-NDC benefited from VRT slightly more than the EG-DC group. Why did this happen? Although the cerebral mechanisms underlying cognitive improvement after virtual rehabilitation are not yet well understood [42], we might hypothesize that repetition over time promotes skill learning because it allows the patients to retain information and knowledge about the quality and effectiveness of their actions, as well as promoting the functional reorganization of damaged cerebral areas and the creation of new cerebral circuits [42,43]. The memory deficits in the EG-DC—especially in Alzheimer’s dementia (63% of our patients)—might make this process more difficult. In this regard, some studies [45,46,47] have highlighted differences between Alzheimer’s and vascular dementia, i.e., patients with Alzheimer’s disease seem to be more affected by memory load and show more impaired retrieval from memory storage, as well as more recall intrusions. On the other hand, prospective and retrospective memory patterns in Alzheimer’s and vascular diseases have been found to be similar by some authors [48]. Future studies could combine VRT with neuroimaging or electroencephalogram, in order to shed light on how brain areas respond to VR stimulation [49]. Cognitive reserve also plays a role in slowing down the clinical progression of dementia [50] and in promoting neuroplasticity mechanisms and the reorganization of an impaired brain, in order to maintain and/or facilitate cognitive performance [51,52]. It might be very useful, for future studies, to include a measure of cognitive reserve within the neuropsychological assessment, in order to identify any differences between dementias of different etiologies, as well as whether there is an association between cognitive reserve and VRT outcomes.

With reference to the second objective of our study—i.e., the spontaneous skill transfer from the virtual environment to the real environment—both EG-DC and EG-NDC increased their correct responses in the natural environment after VRT (intragroup analysis) (Table 2). The total execution time remained generally stable, with the exception of Information and Medicines; therefore, its meaning needs to be clarified, and a comparison with healthy elderly people could be useful for future studies. No significant improvements were found in either CG-DC or CG-NDC in the in vivo test, in the number of correct responses, or in execution time. Since all four groups performed traditional cognitive stimulation, and only the trained groups showed improvements in the in vivo tests, we can hypothesize that traditional cognitive rehabilitation alone is not enough to positively impact daily living skills in the natural environments—at least not in the timeframe of two weeks. 

As already described in the Section 3, the comparisons between the four groups (Kruskal–Wallis test; Table 2) showed statistically significant differences in the correct responses in all of the tasks. After the subsequent post hoc test, better performances were found in the EG-DC than the CG-DC in Information, Suitcase, and Supermarket, as well as in the EG-NDC than in the CG-NDC in Information and Suitcase. These findings seem to confirm that trained groups have better outcomes than untrained groups in the transfer of skills from the laboratory to daily life. However, a comparison is needed between larger groups for future studies; indeed, in the case of Medicines, the statistical significance did not reach the *p*-value set after the Bonferroni correction, and in the Supermarket task no statistically significant differences were found when comparing EG-NDC to EG-NDC.

As in our previous study, the ecological validity of the VRT here seems to be verified, due to the spontaneous transfer of the relearned skills from the virtual setting to the natural life environment. VRT is based on “learning by doing”, which activates reasoning and intuition [53,54]. Through the intuition system, similar characteristics of events can be recognized in different contexts, spontaneously generating a transfer of behaviors from one situation to another. Although the generalization process may be impaired in patients with M-NCDs, as it requires the integrity of cognitive and hippocampal functioning, the results obtained in our study might suggest that VRT is able to promote generalization—at least in the early stages of dementia. 

Regarding the satisfaction questionnaires, moderate-to-high satisfaction was found in both VRT groups, and no problems or adverse events were reported, confirming previous data [5,55]. VRT provides a motivating experience, due to the real scenario to interact with, the immediate feedback, and the enjoyment, which represent critical advantages of the use of VR in rehabilitation compared with traditional paper-and-pencil approaches [42]. Motivation seems to share the same neural network as attention [56], including the cingulate, dorsolateral prefrontal, and inferior parietal cortices; this could explain the compliance and adherence to the VRT generally observed in the patients.

Our study presents some strengths and limitations. Among the strengths, some recommendations by Moreno et al. [55] were taken into account. Therefore, we provided clinical information of the participants, the assessment of satisfaction and adverse effects, the calculation of effect sizes in the case of statistically significant results, and examples of generalization to the real environment. Furthermore, the neuropsychological assessment was carried out with a comprehensive battery, including not only measures on global cognition, but also measures on memory skills and frontal functioning, in order to reduce the eventual influence of any cognitive differences between groups on the results. 

The limitations include the small sample sizes and the prevalence of Alzheimer’s dementia in the DC groups and vascular dementia in the NDC groups. Future studies should include a broader representation of the other etiological categories. We did not provide data on the FLS execution at home, but only in environments arranged at our institute. Due to the lack of follow-up, no information was collected on the maintenance of relearned FLSs over time. Patients were not randomly assigned to the groups, but since our study included a rehabilitative activity, our ethical choice was to provide the VRT to all patients who expressed their acceptance. Finally, our next objective is to increase the sample size in order to be able to use parametric statistics and to thoroughly analyze the effects of the group factor on the time and accuracy variables, under both natural and laboratory conditions. 

## 5. Conclusions

The non-immersive VRT for FLSs produced improvements in the trained daily skills not only in the virtual environment, but also in the natural one, confirming the ecological validity of the VRT in both groups of patients with M-NCD (due to DC and NDC). Despite the clear benefits of using VR for cognitive stimulation and for rehabilitation, virtual systems are still very uncommon in both public and private health services; among the barriers, the complex technical setup may be mentioned, as well as the design and development of 3D interfaces [56]. However, the non-immersive virtual training can be easily repurposed to continue rehabilitation at home (e.g., by using a tablet), because it involves the use of simple technological devices and does not require the constant presence of the trainer. It also causes fewer sickness symptoms (e.g., fatigue, nausea, disorientation, discomfort) and allows patients to maintain control of the surrounding environment. A greater diffusion of virtual reality and a simplification of the virtual technological systems are advisable, as they could significantly improve the quality of healthcare and produce rehabilitation cost savings for families and health services.

## Figures and Tables

**Figure 1 sensors-23-01896-f001:**
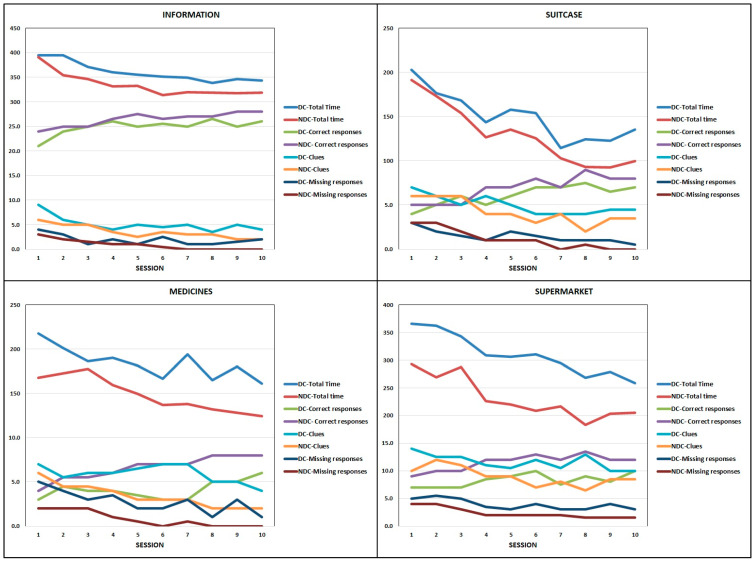
Median number of correct responses, missing responses, and clues, and median values of the total execution time (expressed in s), observed in both degenerative and non-degenerative conditions (DC and NDC) of the EG across the 10 VRT sessions, for each task.

**Table 1 sensors-23-01896-t001:** Comparison of the characteristics of patients with degenerative or non-degenerative conditions in the experimental and control groups at T1. If not otherwise specified, values are expressed as the median (interquartile range). Corrected scores refer to the age- and education-corrected scores.

Sample Characteristics	EG Degenerative Conditions	EG Non-Degenerative Conditions	CG Degenerative Conditions	CG Non-Degenerative Conditions	*p* =
Degenerative/non-degenerative conditions, N	19	21	17	10	NS ^a^
M-NCD severity level, N					NS ^b^
MMSE < 18	4	3	4	1	
MMSE 18–23	13	11	8	6	
MMSE >23	6	13	5	3	
Sex, F/M, N	13/6	12/9	13/4	5/5	NS ^a^
Chronological age, years	71 (65–77.5)	65 (55–71)	69 (62–72)	59.5 (57.25–61.75)	0.015 ^c^*
Education, years	8 (5–10.5)	5 (5–8)	8 (5–13)	8 (8–11)	NS ^c^
Use of technological devices	1 (1–2)	2 (1–2)			NS ^d^
MMSE, corrected score	20.42 (18.61–21.69)	20.29 (19.33–23.14)	19.86 (16–23.86)	20.46 (19.97–23.49)	NS ^c^
CPM, corrected score	20.5 (16.85–24.25)	21.2 (18.4–26.2)	18.9 (15.25–23.75)	20.5 (20.5–30.5)	NS ^c^
Corsi’s test, corrected score	4.51 (3.46–4.66)	3.88 (3.42–4.36)	3.57 (3.09–4.11)	4.07 (3.25–4.15)	NS ^c^
Digit span, corrected score	4.65 (4.15–5.13)	4.34 (3.62–4.65)	4.13 (3.48–5.08)	4.18 (4.11–4.64)	NS ^c^
Rey’s 15 words, immediate recall, corrected score	26.8 (22.18–28.15)	26.9 (22.3–30.9)	20.3 (17.18–28.78)	25.03 (20.03–28.3)	NS ^c^
Rey’s 15 words, delayed recall, corrected score	4.2 (3.15–6.03)	3.1 (0.7–5.7)	2.92 (0–5.83)	2.02 (0.27–6.02)	NS ^c^
FAB, raw score	9 (7–12)	9.5 (8–11)	11 (6–12)	10 (8–13)	NS ^c^
ADL, raw score	6 (6–6)	5.5 (4–6)	5.5 (5–6)	6 (5–6)	NS ^c^
IADL, %	50 (37.5–62.5)	50 (25–62.5)	50 (34.38–75)	80 (63.38–83)	NS ^c^
Information, correct responses	24 (21–24.5)	26 (21–27.25)	24 (22–26)	28 (26.25–28.75)	0.015 ^c+^
Information, total time, s	268 (230–285)	264.5 (243.5–286.5)	293.5 (255.5–361.75)	203 (182.5–225.25)	0.017 ^c”^
Suitcase, correct responses	4 (2–5)	7 (4–9)	5 (2–9)	7.5 (6.25–9)	0.02 ^c°^
Suitcase, total time, s	216 (145–307.5)	238 (179–369)	301.5 (198–439.5)	164 (139–212)	NS ^c^
Medicines, correct responses	6 (4–8)	7 (6–8.5)	5.5 (2.75–7.25)	9 (5.75–10)	NS ^c^
Medicines, total time, s	178 (167–245)	227.5 (140.25–321.75)	236 (164–340)	219 (108–316)	NS ^c^
Supermarket, correct responses	10 (7.5–12)	13 (8–13)	11 (5–12.5)	11 (10–14)	NS ^c^
Supermarket, total time, s	260 (203.5–383)	270 (218–360)	327.5 (252.25–365.25)	247 (202–290)	NS ^c^

EG = Experimental group; CG = control group; M-NCD = major neurocognitive disorder; MMSE = Mini-Mental State Examination; CPM = Colored Progressive Matrices; FAB = Frontal Assessment Battery; ADL = activities of daily living; IADL = instrumental activities of daily living; ^a^ Fisher’s test; ^b^ chi-squared test; ^c^ Kruskal–Wallis test; ^d^ Mann–Whitney U test. Mann–Whitney post hoc test plus Bonferroni correction (*p*-value set to 0.012): * CG degenerative conditions (DG) vs. CG non-degenerative conditions (NDC): z = −2.69, *p* = 0.007; + no statistically significant results in the post hoc test; ” no statistically significant results in the post hoc test; ° EG DC vs. EG NDC: z = 2.53, *p* = 0.01.

**Table 2 sensors-23-01896-t002:** Differences between the second (T3) and first (T1) in vivo tests in the degenerative and non-degenerative conditions of EG and CG, and results of the comparisons between the groups. All values are expressed as the median (interquartile range).

In Vivo Tests	1. EG-DC N = 19	2. EG-NDC N = 21	3. CG-DG N = 17	4. CG-NDC N = 10	Kruskal–Wallis Test	Post Hoc Test *			
						1 vs. 2	3 vs. 4	1 vs. 3	2 vs. 4
	T3-T1	T3-T1	T3-T1	T3-T1	*p* =	z, *p* =	z, *p* =	z, *p* =	z, *p* =
Information correct responses	3 (2/3.5) **	3 (1.75–4.25) **	0 (−1/1)	0 (−0.75/0)	0.00006	NS	NS	3.3 0.00094	3.3 0.0096
Information total time, s	–38 (–73.5/–19.5) **	–62.5 (–108.5/–48) **	−48 (−85/−11) **	−17.5 (−42.25/0)	NS				
Suitcase correct responses	3 (0.5–5) **	2 (0–4) **	0 (−1/1)	0 (−1.75/−0.75)	0.0009	NS	NS	2.8 0.005	2.85 0.004
Suitcase total time, s	6 (–41/67)	–41 (–137/16)	−16 (−103/−4)	−21.5 (−28.75/60.75)	NS				
Medicines correct responses	2 (0/3) **	1 (0.5/2.5)	0 (0/2)	0 (0/0)	0.016	NS	NS	2.22 0.03	2.29 0.02
Medicines total time, s	–39 (–80/–4)	–57 (–91/–22.5) **	1.5 (−71.25/27.5)	−8 (−53.5/−3)	NS				
Supermarket correct responses	2 (1/4) **	1 (1/4) **	0 (0/1)	1 (0/1)	0.037	NS	NS	2.6 0.009	NS
Supermarket total time, s	30 (–145/72.5)	–27 (–109/54)	−31 (−61/−2)	−7 (−55.5/3.75)	NS				

EG = experimental group; CG = control group; NC = degenerative conditions; NDC = non-degenerative conditions; CG = control group; * post hoc test: Mann–Whitney U test + Bonferroni correction (*p*-value set to 0.0125); ** statistically significant intragroup difference (Wilcoxon matched-pairs test + Bonferroni correction; *p*-value set to 0.006).

**Table 3 sensors-23-01896-t003:** Results obtained during the 10 VRT sessions from the EGs with both degenerative and non-degenerative conditions (Friedman test for repeated measures; all values are expressed as *p*-values; Kendall’s W coefficient of concordance *).

	Correct Responses	Errors	Missing Responses	Clues	Total Time, s
Non-degenerative conditions					
Information	0.00001; 0.22	NS	<0.00001; 0.37	0.00001; 0.22	<0.00001; 0.4
Suitcase	<0.00001; 0.39	NS	<0.00001; 0.38	<0.00001; 0.37	<0.00001; 0.54
Medicines	0.00037; 0.18	NS	<0.00001; 0.25	0.00037; 0.18	<0.00001; 0.46
Supermarket	<0.00001; 0.32	0.0085; 0.12	<0.00001; 0.24	<0.00001; 0.31	<0.00001; 0.33
Degenerative conditions					
Information	0.00001; 0.0.24	NS	<0.00001; 0.0.29	0.00001; 0.25	<0.00001; 0.56
Suitcase	0.002; 0.18	NS	0.008; 0.15	0.0007; 0.15	0.00001; 0.28
Medicines	0.03; 0.13	NS	<0.00001; 0.24	0.027; 0.13	<0.00001; 0.31
Supermarket	0.025; 0.07	NS	0.00008; 0.24	0.00001; 0.27	<0.00001; 0.31

* The Kendall’s W coefficient of concordance uses the Cohen’s interpretation guidelines of 0.1 (small effect), 0.3 (moderate effect), and 0.5 (strong effect).

**Table 4 sensors-23-01896-t004:** Results of the comparisons between the EG—degenerative and non-degenerative conditions—in the 1st, 5th, and 10th sessions.

	1st Session	5th Session	10th Session
Information, correct responses	z = 1.29; *p* = 0.2	z = 1.7; *p* = 0.09	z = 1.91; *p* = 0.06
Information, missing responses	z = 0.66; *p* = 0.51	z = 1.22; *p* = 0.22	z = 2.17; *p* = 0.03; r = 0.35
Information, clues	z = −1.3; *p* = 0.19	z = −1.67; *p* = 0.09	z = −1.91; *p* = 0.06
Information, total time, s	z = 0.04; *p* = 0.97	z = 0.87; *p* = 0.38	z = 0.91; *p* = 0.36
Suitcase, correct responses	z = 1.04; *p* = 0.3	z = 1.18; *p* = 0.24	z = 2.09; *p* = 0.037; r = 0.33
Suitcase, missing responses	z = 0.7; *p* = 0.48	z = 1.62; *p* = 0.1	z = 1.58; *p* = 0.11
Suitcase, clues	z = 0.64; *p* = 0.52	z = 0.74; *p* = 0.46	z = 1.71; *p* = 0.09
Suitcase, total time, s	z = 0.25; *p* = 0.8	z = 0.32; *p* = 0.75	z = 1.29; *p* = 0.19
Medicines, correct responses	z = 1.33; *p* = 0.18	z = 2.28; *p* = 0.023; r = 0.37	z = 0.87; *p* = 0.38
Medicines, missing responses	z = 2.31; *p* = 0.021; r = 0.37	z = 1.68; *p* = 0.093	z = 1.09; *p* = 0.28
Medicines, clues	z = 1.33; *p* = 0.18	z = 2.28; *p* = 0.023; r = 0.37	z = 0.87; *p* = 0.38
Medicines, total time, s	z = 0.99; *p* = 0.32	z = 0.98; *p* = 0.33	z = 0.52; *p* = 0.6
Supermarket, correct responses	z = 1.59; *p* = 0.11	z = 2.97; *p* = 0.003; r = 0.48	z = 2.12; *p* = 0.034; r = 0.34
Supermarket, missing response	z = 1.55; *p* = 0.25	z = 2.04; *p* = 0.04; r = 0.33	z = 1.48; *p* = 0.14
Supermarket, clues	z = 1.59; *p* = 0.11	z = 1.15; *p* = 0.25	z = 1.28; *p* = 0.2
Supermarket, total time, s	z = 0.23; *p* = 0.8	z = 0.81; *p* = 0.42	z = 1.58; *p* = 0.11

## Data Availability

The data presented in this study are openly available in Mendeley at 10.17632/ygphd2gfvb.1.

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
