# Peer review of "Functional Living Skills in Patients with Major Neurocognitive Disorder Due to Degenerative or Non-Degenerative Conditions: Effectiveness of a Non-Immersive Virtual Reality Training"

_sensors, 2023, doi:10.3390/s23041896_

Round 1
Reviewer 1 Report
The study is interesting, however the rationale of the study is improveble and in particular several references are missing at all. The Authors should consider that there is a huge scientific literature on neuropsychology and vr and in particular executive functions and vr and other speficific functioning.
As highlighted in a recent study (https://www.frontiersin.org/articles/10.3389/fpsyg.2018.02086/full), most of recent VR publications are related to clinical issues and this should be highlighted.
The methods are correct but need to be defined in a clearer way. Why the Authors have chosen non parametric analysis instead of rmANOVA to study interaction effects? Moreover if the Authors have specific hypothesis on similitude among groups or timepoints, they should use Bayes factor to test this instead to highlight non significant results. With so many time point treatment the Authors could have used panel data methods or hierarchical multilevel regression, or similar. The should explain or include this in the limitations of the study. Another thing: did Authors correct p-value after Fisher test, to correct error? They should, at least, divide for the possible couple to compare (3 timepoints means [(3x2)/2]=3 couple comparisons, 4 timepoints means [(4x3)/2]=6 couple comparisons), so that an alpha of .05 might be corrected to .05/3 or .05/6 and this changes the results. Please control.
Drafted conclusions makes sense, however should be strictly connected with a more clear rationale to be defined in the introduction.
Author Response
Reply to reviewer 1
Dear reviewer 1, thank you for your suggestions. Below in red the replies.
English very difficult to understand/incomprehensible
R: Thanks for your comment. We have tried to check the manuscript thoroughly in order to make the complex content more easy to be followed and the English form has been checked by a skilled English translator. We believe that the content of the paper is rather complex but the English form is now basically appropriate.
The study is interesting, however the rationale of the study is improveble and in particular several references are missing at all. The Authors should consider that there is a huge scientific literature on neuropsychology and vr and in particular executive functions and vr and other specific functioning.
As highlighted in a recent study (https://www.frontiersin.org/articles/10.3389/fpsyg.2018.02086/full), most of recent VR publications are related to clinical issues and this should be highlighted.
R: In the Introduction we add four studies, included the article suggested by the reviewer, highlighting the clinical use of VR, and the use of VR for investigating the executive functioning, especially in everyday situations (lines 49-63).
The studies are:
- Cipresso, P.; Giglioli, I. A. C.; Raya, M. A.; Riva, G. The Past, Present, and Future of Virtual and Augmented Reality Research: A Network and Cluster Analysis of the Literature. Psychol. 2018, 9, 2086, doi:10.3389/fpsyg.2018.02086.
- Parsons, T. D.; Gaggioli, A.; Riva, G. Extended reality for the clinical, affective, and social neurosciences. Brain Sci. 2020, 10, 1–22, doi: 10.3390/brainsci10120922.
- Kim, E.; Han, J.; Choi, H.; Prié, Y.; Vigier, T.; Bulteau, S.; Kwon, G. H. Examining the Academic Trends in Neuropsychological Tests for Executive Functions Using Virtual Reality: Systematic Literature Review. JMIR serious games, 2021, 9, e30249, doi:10.2196/30249.
- Tomaszewski Farias, S.; Cahn-Weiner, D. A.; Harvey, D.J.; Reed, B. R.; Mungas, D.; Kramer, J. H. ; Chui, H. Longitudinal changes in memory and executive functioning are associated with longitudinal change in instrumental activities of daily living in older adults. Clin. Neuropsychol. 2009, 23, 446-461, doi:10.1080/13854040802360558.
The methods are correct but need to be defined in a clearer way.
Why the Authors have chosen non parametric analysis instead of rmANOVA to study interaction effects?
Moreover if the Authors have specific hypothesis on similitude among groups or timepoints, they should use Bayes factor to test this instead to highlight non significant results.
With so many time point treatment the Authors could have used panel data methods or hierarchical multilevel regression, or similar.
The should explain or include this in the limitations of the study.
R: we chose to use nonparametric tests because the groups to be compared did not show a normal distribution and had a small size. However, our objective is to expand the sample size (patients are constantly recruited) and to be able to use parametric tests, such as the Manova, to thoroughly analyze the effects of the group factor on time and accuracy variables, under natural and laboratory conditions. In the discussion section we explicitly included this concept.
Another thing: did Authors correct p-value after Fisher test, to correct error? They should, at least, divide for the possible couple to compare (3 timepoints means [(3x2)/2]=3 couple comparisons, 4 timepoints means [(4x3)/2]=6 couple comparisons), so that an alpha of .05 might be corrected to .05/3 or .05/6 and this changes the results. Please control.
R: We utilized the chi-square test in three cases (degenerative and non-degenerative conditions, severity levels of dementia, and number of M ad F). With regard to degenerative or non-degenerative conditions, and to the number of F and M, after your comment we replaced the chi-square test with the Fisher test, with a 2X2 contingency table: in the first case, EG and CG x two variables, such as DC e NDC; in the second case DC and NDC x two variables, such as F and M. Results were not statistically significant (table 1).
Drafted conclusions makes sense, however should be strictly connected with a more clear rationale to be defined in the introduction.
R: the Introduction has been expanded, and the objectives clarified better. The Discussion has been modified in some parts, and punctually referred to the objectives of the study.
Reviewer 2 Report
This paper aimed to compare the effectiveness of a low-immersive Virtual Reality training on four functional living skills in patients with Major Neurocognitive Disorders due to degenerative or non-degenerative conditions. The paper is well structured, and generally well written.
Overall the manuscript is well written. I have a few concerns/suggestions.
Abstract
the meaning of the acronym M-NCDs must be added.
Introduction
The introduction is well written. I have no concerns about this section.
Materials and Methods / Results
Tables must be reorganized. In particular, I suggest dividing the control group into two subgroups (Degenerative/Non-degenerative conditions), as was done for the experimental group. I also suggest clarifying in more detail how the statistical analysis was carried out and what the results in the table refer to, in particular between which groups the statistical analysis was carried out and how the results are shown in the table.
Discussion
The discussion is well written. both future developments and work limitations are included. I suggest referring to other studies that have achieved good results using similar technology such as:
Maranesi E, et al. The Effect of Non-Immersive Virtual Reality Exergames versus Traditional Physiotherapy in Parkinson's Disease Older Patients: Preliminary Results from a Randomized-Controlled Trial. Int J Environ Res Public Health. 2022 Nov 10;19(22):14818. doi: 10.3390/ijerph192214818.
Author Response
Reply to reviewer 2
Dear reviewer 2, thank you for your suggestions. Below in red the replies.
This paper aimed to compare the effectiveness of a low-immersive Virtual Reality training on four functional living skills in patients with Major Neurocognitive Disorders due to degenerative or non-degenerative conditions. The paper is well structured, and generally well written.
Overall the manuscript is well written. I have a few concerns/suggestions.
Abstract
The meaning of the acronym M-NCDs must be added.
R: done. Moreover, the abstract was generally revised since we divided the control group into two subgroups (DC and NDC). Now, the groups were indicated as EG-DC (experimental group – degenerative conditions), EG-NDC (experimental group – non-degenerative conditions), CG-DC (control group – degenerative conditions) and CG-NDC (control group – non-degenerative conditions).
Introduction
The introduction is well written. I have no concerns about this section.
Materials and Methods / Results
Tables must be reorganized. In particular, I suggest dividing the control group into two subgroups (Degenerative/Non-degenerative conditions), as was done for the experimental group.
R: tables 1 and 2 were reorganized, the control group was divided into two subgroups (CG-DC and CG-NDC), the statistical analysis was redone, and results were included in the tables. The new results were discussed in the Discussion section
I also suggest clarifying in more detail how the statistical analysis was carried out and what the results in the table refer to, in particular between which groups the statistical analysis was carried out and how the results are shown in the table.
R: the 2.4. Statistical analysis was detailed more, as follows:
Recorded data did not show a normal distribution shape (asymmetry and kurtosis calculations). For this reason, and since the groups were small in number, nonparametric statistics were used. The comparisons between the EG (DC and NDC) and CG (DC and NDC) groups was carried out by means of the Fisher test for dementia etiology and sex (2x2 contingency table), Chi square test for severity level of dementia (4 x 3 contingency table). All the other comparisons between the four groups were done by the Kruskal-Wallis test. When a statistically significant difference was found, a post-hoc test was conducted with the Mann-Whitney’s U test and Bonferroni correction (p value was set to 0.012). For the in vivo tests, the within group comparisons were performed by using the Wilcoxon matched pairs test followed by the Bonferroni correction (setting the p value to 0.006). The Friedman test for repeated measures was used in order to compare the 10 VRT sessions carried out with the EG (both DC and NDC), together with the Kendall’s W coefficient of concordance. In order to assess the differences in the session progression between EG DC and EG NDC, the 1st, the 5th, and the 10th sessions were compared by means of the Mann-Whitney U test. The r effect sizes were also calculated (r = z/√N; 0.1= small effect size; 0.3=medium effect size; 0.5 = large effect size).
Discussion
The discussion is well written. both future developments and work limitations are included. I suggest referring to other studies that have achieved good results using similar technology such as:
Maranesi E, et al. The Effect of Non-Immersive Virtual Reality Exergames versus Traditional Physiotherapy in Parkinson's Disease Older Patients: Preliminary Results from a Randomized-Controlled Trial. Int J Environ Res Public Health. 2022 Nov 10;19(22):14818. doi: 10.3390/ijerph192214818.
R: thank you very much, we added the reference suggested in the Introduction section (lines 96-101).
